# Reverse engineering learned optimizers reveals known and novel mechanisms

**Niru Maheswaranathan**[*]
Google Research, Brain Team
niru@hey.com

**David Sussillo**[*]
Google Research, Brain Team

**Luke Metz**
Google Research, Brain Team
lmetz@google.com

**Ruoxi Sun**
Google Research, Brain Team
ruoxis@google.com

**Jascha Sohl-Dickstein**
Google Research, Brain Team
jaschasd@google.com

## Abstract

Learned optimizers are parametric algorithms that can themselves be trained to solve optimization problems. In contrast to baseline optimizers (such as momentum or Adam) that use simple update rules derived from theoretical principles, learned optimizers use flexible, high-dimensional, nonlinear parameterizations. Although this can lead to better performance, their inner workings remain a mystery. How is a given learned optimizer able to outperform a well tuned baseline? Has it learned a sophisticated combination of existing optimization techniques, or is it implementing completely new behavior? In this work, we address these questions by careful analysis and visualization of learned optimizers. We study learned optimizers trained from scratch on four disparate tasks, and discover that they have learned interpretable behavior, including: momentum, gradient clipping, learning rate schedules, and learning rate adaptation. Moreover, we show how dynamics and mechanisms inside of learned optimizers orchestrate these computations. Our results help elucidate the previously murky understanding of how learned optimizers work, and establish tools for interpreting future learned optimizers.

## 1 Introduction

Optimization algorithms underlie nearly all of modern machine learning; thus advances in optimization have broad impact. Recent research uses meta-learning to learn new optimization algorithms, by directly parameterizing and training an optimizer on a distribution of tasks. These so-called *learned optimizers* have been shown to outperform baseline optimizers in restricted settings [1–7].

Despite improvements in the design, training, and performance of learned optimizers, fundamental questions remain about their behavior. We understand remarkably little about *how* these optimizers work. Are learned optimizers simply learning a clever combination of known techniques? Or do they learn fundamentally new behaviors that have not yet been proposed in the optimization literature? If they did learn a new optimization technique, how would we know?

Contrast this with existing "hand-designed" optimizers such as momentum [8], AdaGrad [9], RMSProp [10], or Adam [11]. These algorithms are motivated and analyzed using intuitive mechanisms and theoretical principles (such as accumulating update velocity in momentum, or rescaling updates based on gradient magnitudes in RMSProp or Adam). This understanding of underlying mechanisms allows future studies to build on these techniques by highlighting flaws in their operation [12],

---

[*]Work conducted while at Google Research. Currently at Meta Reality Labs.

35th Conference on Neural Information Processing Systems (NeurIPS 2021).

studying convergence [13], and developing deeper knowledge about why key mechanisms work [14]. Without analogous understanding of the inner workings of a learned optimizers, it is incredibly difficult to analyze or synthesize their behavior.

In this work, we develop tools for isolating and elucidating mechanisms in nonlinear, high-dimensional learned optimization algorithms (§4). Using these methods we show how learned optimizers utilize both known and novel techniques, across four disparate tasks. In particular, we demonstrate that learned optimizers learn momentum (§5.1), gradient clipping (§5.2), learning rate schedules (§5.3), and methods for learning rate adaptation (§5.4, §5.5). Taken together, our work can be seen as part of a new approach to scientifically interpret and understand learned algorithms.

We provide code for training and analyzing learned optimizers, as well as the trained weights for the learned optimizers studied here, at `https://bit.ly/3eqgNrH`.

## 2 Related Work

Our work is heavily inspired by recent work using neural networks to parameterize optimizers. Andrychowicz et al. [1] originally showed promising results on this front, with additional studies improving robustness [2, 3], meta-training [6], and generalization [7] of learned optimizers.

Here, we study the behavior of optimizers by treating them as dynamical systems. This perspective has yielded a number of intuitive and theoretical insights [15–18]. We also build on recent work on reverse engineering recurrent neural networks (RNNs). Sussillo and Barak [19] showed how linear approximations of nonlinear RNNs can reveal the algorithms used by trained networks to solve simple tasks. These techniques have been applied to understand trained RNNs in a variety of domains, from natural langauge processing [20, 21] to neuroscience [22]. Additional work on treating RNNs as dynamical systems has led to insights into their computational capabilities [23–25].

## 3 Methods

### 3.1 Preliminaries

We are interested in optimization problems that minimize a loss function ($f$) over parameters ($\boldsymbol{x}$). We focus on first-order optimizers, which at iteration $k$ have access to the gradient $g_i^k \equiv \nabla f(x_i^k)$ and produce an update $\Delta x_i^k$. These are *component-wise* optimizers that are applied to each parameter ($x_i$) of the problem in parallel. Standard optimizers used in machine learning (e.g. momentum, Adam) are in this category[1]. Going forward, we use $x$ for the parameter to optimize, $g$ for its gradient, $k$ for the current iteration, and drop the parameter index ($i$) to reduce excess notation.

One can think of an optimizer as being comprised of two parts: the optimizer state ($\boldsymbol{h}$) that stores information about the current problem, and readout weights ($\boldsymbol{w}$) that are used to update parameters given the current state. An optimization algorithm is specified by an initial state, state transition dynamics, and readout, defined as follows:

$$
\begin{aligned}
\boldsymbol{h}^{k+1} &= F(\boldsymbol{h}^k, g^k) & (1) \\
x^{k+1} &= x^k + \boldsymbol{w}^T \boldsymbol{h}^{k+1}, & (2)
\end{aligned}
$$

where $\boldsymbol{h}$ is the optimizer state, $F$ governs the optimizer dynamics, and $\boldsymbol{w}$ are the readout weights.

*Learned optimizers* are constructed by parameterizing the function $F$, and then learning those parameters along with the readout weights through meta-optimization (detailed in App. D.2). *Hand-designed* optimization algorithms, by distinction, specify these functions at the outset.

For example, in momentum, the state variable is a single number (known as the velocity), and is updated as a linear combination of the previous state and the gradient (e.g. $h^{k+1} = \beta h^k + g^k$, where $\beta$ is the momentum hyperparameter). For momentum and other hand-designed optimizers, the state variables are low-dimensional, and their dynamics are (largely) straightforward. In contrast, learned optimizers have high-dimensional state variables, and the potential for rich, nonlinear dynamics. As these systems learn complex behaviors, it has historically been difficult to extract simple, intuitive descriptions of the behavior of a learned optimizer.

---

[1]Notable exceptions include quasi-Newton methods such as L-BFGS [26] or K-FAC [27].

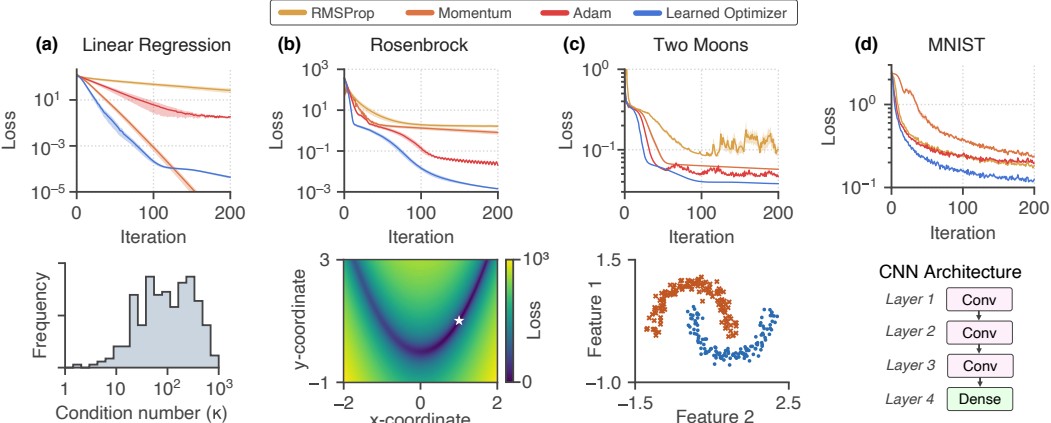

Figure 1: Learned optimizers outperform well tuned baselines on four tasks: **(a)** linear regression, **(b)** the Rosenbrock function, **(c)** training a fully connected neural network on the two moons dataset, and **(d)** training a convolutional neural network on the MNIST dataset. *Top row*: Optimizer performance, shown as loss curves (mean $\pm$ std. error over 64 random seeds) for momentum (orange), RMSProp (yellow), Adam (red) and a learned optimizer (blue). *Bottom row*: Additional information pertaining to each task (described in §3.2).

## 3.2 Training learned optimizers

We parametrize the learned optimizer with a recurrent neural network (RNN), similar to Andrychowicz et al. [1]. The only input to the optimizer is the gradient. The RNN is trained by minimizing a meta-objective, which we define as the average training loss when optimizing a target problem. See App. D.2 for details about the optimizer architecture and meta-training procedures. Below, we only analyze the final (best) trained optimizer, however we do analyze aspects of the meta-training dynamics in App B.2.

We trained learned optimizers on each of four tasks. These tasks were selected because they converge in a relatively small number of iterations (particularly important for meta-optimization) and cover a range of loss surfaces (convex and non-convex, low- and high-dimensional, deterministic and stochastic):

**Linear regression:** The first task consists of random linear regression problems $f(\boldsymbol{x}) = \frac{1}{2}\|\boldsymbol{A}\boldsymbol{x}-\boldsymbol{b}\|_2^2$, where $\boldsymbol{A}$ and $\boldsymbol{b}$ are randomly sampled. Much of our theoretical understanding of the behavior of optimization algorithms is derived using quadratic loss surfaces such as these, in part because they have a constant Hessian ($\boldsymbol{A}^T\boldsymbol{A}$) over the entire parameter space. The choice of how to sample the problem data $\boldsymbol{A}$ and $\boldsymbol{b}$ will generate a particular distribution of Hessians and condition numbers. A histogram of condition numbers for our task distribution is shown in Figure 1a.

**Rosenbrock:** The second task is minimizing the Rosenbrock function [28], a commonly used test function for optimization. It is a non-convex function with a curved valley and a single global minimum. The function is defined over two parameters as $f(x, y) = (1 - x)^2 + 100(y - x^2)^2$ (Figure 1b). The distribution of problems for this task consists of different initializations sampled uniformly over a grid. The grid used to sample initializations is the same as the grid shown in the figure; the x- and y- coordinates are sampled from the ranges (-2, 2) and (-1, 3), respectively.

**Two moons:** The third task involves training a fully connected neural network to classify a toy dataset, the two moons dataset (Figure 1c). As the data are not linearly separable, a nonlinear classifier is required. The optimization problem is to train the weights of a three hidden layer fully connected neural network, with 64 units per layer and tanh nonlinearities (for a total of 8,577 parameters). The distribution of problems involves sampling the initial weights of the fully connected network.

**MNIST:** The fourth task is to train a four layer convolutional network to classify digits from the MNIST dataset. We use a minibatch size of 100 examples; thus the gradients fed to the optimizer are stochastic, unlike the previous three problems. The network consists of three convolutional

layers each with 16 channels with a 3×3 kernel size and ReLU activations, followed by a final (fully connected) dense layer, for a total of 82,250 parameters (Figure 1d).

We additionally tuned three baseline optimizers (momentum, RMSProp, and Adam) individually for each task. We selected the hyperparameters for each problem out of 2500 samples randomly drawn from a grid. Details about the exact grid ranges used for each task are in App. D.3.

Figure 1 (top row) compares the performance of the learned optimizer (blue) to baseline optimizers (red, yellow, and orange), on each of the four tasks described above. Across all tasks, the learned optimizer outperforms the baseline optimizers on the meta-objective[2] (App. Fig. 17).

# 4 Tools for understanding optimizers

In order to analyze learned optimizers, we make extensive use of two methods. The first is a way to visualize what an optimizer is doing at a particular optimizer state. The second is a way of making sense of how the optimizer state changes, that is, understanding the optimizer state dynamics. We describe these below, and then use them to analyze optimizer behavior and mechanisms in §5.

## 4.1 Update functions

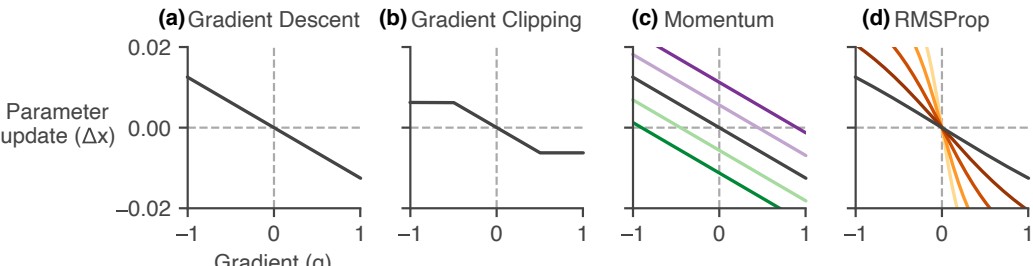

Figure 2: Visualizing optimizer behavior with update functions (see §4.1 for details) for different commonly used optimization techniques. **(a)** Gradient descent is a (stateless) linear function, whose slope is the learning rate. **(b)** Gradient clipping saturates the update, beyond a threshold. **(c)** Momentum introduces a vertical offset depending on the accumulated velocity (colors indicate different values of the accumulated momentum). **(d)** RMSProp changes the slope (effective learning rate) of the update (colors denote changes in the state variable, the accumulated squared gradient).

First, we introduce a visualization tool to get a handle on what an optimizer is doing. Any optimizer, at a particular state, can be viewed as a scalar function that takes in a gradient ($g$) and returns a change in the parameter ($\Delta x$). We refer to this as the optimizer *update function*.

Mathematically, the update function is computed as the state update projected onto the readout, $\Delta x = \boldsymbol{w}^T F(\boldsymbol{h}, g)$, following equations (1) and (2). In addition, the slope of this function with respect to the input gradient $\left( \frac{\partial \Delta x}{\partial g} \right)$ can be thought of as the *effective learning rate* at a particular optimizer state[3]. A steeper slope means that the parameter update is more sensitive to the gradient (as expected for a higher learning rate), and a shallower slope means that parameter updates are smaller for the same gradient magnitude (i.e. a lower learning rate).

As the optimizer state varies, the update function and effective learning rate can change. This provides a mechanism for learned optimizers to implement different types of behavior: through optimizer state dynamics that induce particular types of changes in the update function.

---

[2]As the meta-objective is the average training loss during an optimization run, it naturally penalizes the training curve earlier in training (when loss values are large). This explains the discrepancy in the curves for linear regression (Fig. 1a, top) where momentum continues to decrease the loss late in training. Despite this, the learned optimizer has an overall smaller meta-objective due to having lower loss at earlier iterations.

[3]We compute this slope at $g = 0$, in the middle of the update function. We find that the update function is always affine in the middle with saturation at the extremes, thus the slope at $g = 0$ is a natural way to summarize the effective learning rate.

It is instructive to first visualize update functions for commonly used optimizers (Figure 2). For gradient descent, the update ($\Delta x = -\alpha g$) is stateless and is always a fixed linear function whose slope is the learning rate, $\alpha$ (Fig. 2a). Gradient clipping is also stateless, but is a saturating function of the gradient (Fig. 2b). For momentum, the update is $\Delta x = \beta v - \alpha g$, where $v$ denotes the momentum state (velocity) and $\beta$ is the momentum timescale. The velocity adds an offset to the update function (Fig. 2c). As the optimizer picks up positive (or negative) momentum, the curve shifts downward (or upward), thus incorporating a bias to reduce (or increase) the parameter. For adaptive optimizers such as RMSProp, the state variable changes the slope, or effective learning rate, within the linear region of the update function (Fig. 2d).

Now, what about learned optimizers, or optimizers with much more complicated or high-dimensional state variables? One advantage of update functions is that, as scalar functions, they can be easily visualized and compared to the known methods in Figure 2. Whether or not the underlying hidden states are interpretable, for a given learned optimizer, remains to be seen.

### 4.2 A dynamical systems perspective

In order to understand the state dynamics of a learned optimizer, we approximate the nonlinear dynamical system (eq. (1)) via linearized approximations [29]. These linear approximations hold near *fixed points* of the dynamics. Fixed points are points in the state space of the optimizer, where — as long as input gradients do not perturb it — the system does not move. That is, an approximate fixed point $\boldsymbol{h}^*$ satisfies the following: $\boldsymbol{h}^* \approx F(\boldsymbol{h}^*, g^*)$, for a particular input $g^*$.

We numerically find approximate fixed points [19, 30] by solving an optimization problem where we find points ($\boldsymbol{h}$) that minimize the following loss: $\frac{1}{2}\|F(\boldsymbol{h}, g^*) - \boldsymbol{h}\|_2^2$. The solutions to this problem (there may be many) are approximate fixed points of the system $F$ for a given input, $g^*$. In general, there may be different fixed points for different values of the input ($g$). First we will analyze fixed points when $g^* = 0$ (§5.1), and then later discuss additional behavior that occurs as $g^*$ varies (§5.4).

One can think of the structure of fixed points as shaping a dynamical skeleton that governs the optimizer behavior. As we will see, for a well trained optimizer, the dynamics around fixed points enable interesting and useful computations.

## 5 Mechanisms of learned optimizers

We selected and analyzed the best learned optimizer on each of the four tasks in §3.2. Across these, we discovered a number of mechanisms responsible for their superior performance: momentum, gradient clipping, learning rate schedules, and new types of learning rate adaptation In the following sections, we go through each mechanism in detail, showing how it is implemented.

In general, we found similar mechanisms across learned optimizers trained on all four tasks. Thus, for brevity, we only show the results in the main text from one optimizer for each mechanism. For any mechanisms that were found to be task dependent, we point this out in the relevant section. Results for all tasks are presented in App. A.

### 5.1 Momentum

We discovered that learned optimizers implement classical momentum, and do so using dynamics that are well described by a linear approximation.

To see how, first consider the dynamics of a learned optimizer near a fixed point. Here, we can linearly approximate the state dynamics (eq. 1) using the Jacobian of the optimizer update [29, 19]. The *linearized* state update is given by

$$F(\boldsymbol{h}^k, g^k) \approx \boldsymbol{h}^* + \frac{\partial F}{\partial \boldsymbol{h}}\left(\boldsymbol{h}^k - \boldsymbol{h}^*\right) + \frac{\partial F}{\partial g}g^k, \tag{3}$$

where $\boldsymbol{h}^*$ is a fixed point of the dynamics, $\frac{\partial F}{\partial \boldsymbol{h}}$ is the Jacobian matrix, and $\frac{\partial F}{\partial g}$ is a vector that controls how the scalar gradient enters the system. Both of these latter two quantities are evaluated at the fixed point, $\boldsymbol{h}^*$, and $g^* = 0$.

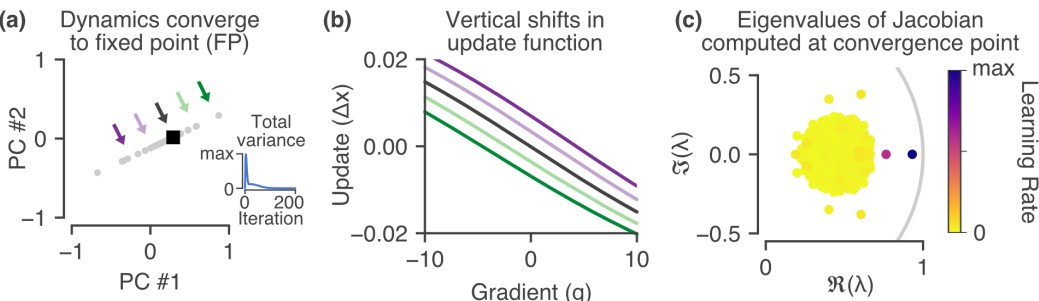

Figure 3: Momentum in learned optimizers. Plots are for the optimizer trained on the Rosenbrock task (we see similar behavior for optimizers trained on the other tasks, see App. A.1). **(a)**: Projection of the optimizer state near a convergence point (black square). *Inset:* the total variance of the optimizer states over test problems goes to zero as the trajectories converge. **(b)**: visualization of the update functions (§4.1) along the slow mode of the dynamics (colored lines correspond to arrows in (a)). Along this dimension, the effect on the system is to induce an offset in the update, just as in classical momentum (cf. Fig. 2c). **(c)**: Eigenvalues of the linearized optimizer dynamics at the fixed point (black square in (a)) plotted in the complex plane. The eigenvalue magnitudes are momentum timescales, and the color indicates the corresponding learning rate. See §5.1 for details.

This Jacobian is a matrix with $N$ eigenvalues and eigenvectors, where $N$ is the dimensionality of the optimizer state (for the RNN architectures that we use, $N$ is the number of RNN units). For a linear dynamical system, as we have now, the dynamics decouple along the $N$ eigenmodes of the system.

Writing the update along these coordinates (let's denote them as $v_j$, for $j = 1, \ldots, N$) allows us to rewrite the learned optimizer as a momentum algorithm (see App. C for details) with $N$ timescales:

$$v_j^{k+1} = \beta_j v_j^k + \alpha_j g + \text{const.},$$

where the magnitude of the eigenvalues of the Jacobian are exactly momentum timescales ($\beta_j$), each with a corresponding learning rate ($\alpha_j$). Incidentally, momentum with multiple timescales has been previously studied and called aggregated momentum by Lucas et al. [31].

To summarize, near a fixed point, a (nonlinear) learned optimizer is approximately linear, and the eigenvalues of the its Jacobian can be interpreted as momentum timescales.

We then looked to see if (and when) learned optimizers operated near fixed points. We found that across all tasks, optimizers converged to fixed points; often to a single fixed point. Figure 3 shows this for a learned optimizer trained on the Rosenbrock task. Fig. 3a is a 2D projection of the hidden state (using principal components analysis[4]), and shows the single fixed point for this optimizer (black square). All optimizer state trajectories converge to this fixed point, this can be seen as the total variance of optimizer states across test examples goes to zero (Fig. 3a, inset).

Around this fixed point, the dynamics are organized along a line (gray circles). Shifting the hidden state along this line (indicated by colored arrows) induces a corresponding shift in the update function (Fig. 3b), similar to what is observed in classical momentum (cf. Fig. 2c).

This learned optimizer uses a single eigenmode to implement momentum. Fig. 3c shows the eigenvalues of the Jacobian (computed at the convergence fixed point) in the complex plane, colored by that mode's learning rate (see App. C for how these quantities are computed). This reveals a single dominant eigenmode (colored in purple), whose eigenvector corresponds to the momentum direction (gray points in Fig. 3a) and whose eigenvalue is the corresponding momentum timescale.

For some learned optimizers, this momentum eigenvalue exactly matched the best tuned momentum hyperparameter; in these cases the optimizer's performance matched that of momentum as well. We analyze one such optimizer in App. B as it is instructive for understanding the momentum mechanism.

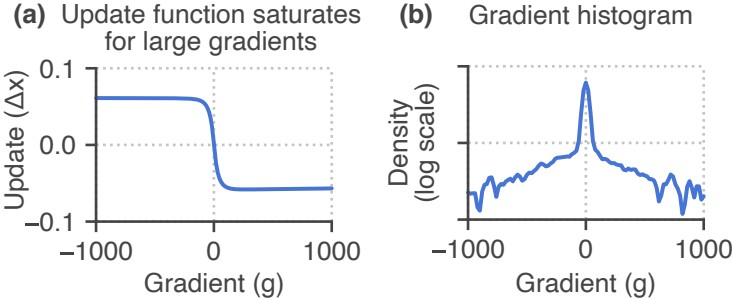

Figure 4: Gradient clipping in a learned optimizer trained on the Rosenbrock task (results for additional tasks are in App. A.2). **(a)**: The update function computed at the initial state saturates for large gradient magnitudes. The effect of this is similar to that of gradient clipping (cf. Fig. 2b). **(b)**: The empirical density of encountered gradients for this task. This shows that while most of the gradients occur in the linear regime, a small but non-negligible fraction are quite large and will saturate the update function.

## 5.2  Gradient clipping

In standard gradient descent, the parameter update is a linear function of the gradient. Gradient clipping [32] instead modifies the update to be a saturating function (Fig. 2b). This prevents large gradients from inducing large parameter changes, which is useful for optimization problems with non-smooth gradients [14].

We find that learned optimizers also use saturating update functions as the gradient magnitude increases, thus learning a soft form of gradient clipping. We show this for the learned optimizer trained on the Rosenbrock problem in Figure 4a. Although Fig. 4a shows the saturation for a particular optimizer state (the initial state in this case), we find that these saturating thresholds are consistent throughout the optimizer state space.

The strength of the clipping effect depends on the training task. We can see this by comparing the update function for a given optimizer to the distribution of gradients encountered for that task (Fig. 4b); the higher the probability of encountering a gradient that is in the saturating regime of the update function, the more clipping is used.

For some problems, such as linear regression, the learned optimizer largely stays within the linear region of the update function (App. A.2). For others, such as the Rosenbrock problem presented in Fig. 4, the optimizer utilizes more of the saturating part of the update function.

## 5.3  Learning rate schedules

Practitioners often tune learning rate schedules along with other optimization hyperparameters. Originally motivated to guarantee convergence in stochastic optimization [33], schedules are now used more broadly [34–37]. These schedules are typically a decaying function of the iteration — meaning the learning rate goes down as optimization progresses — although Goyal et al. [38] use an additional increasing warm-up period, and even more exotic schedules have been proposed [39–41].

We discovered that learned optimizers can implement a schedule using *autonomous* — that is, not input driven — dynamics. If the initial optimizer state is away from fixed points of the state dynamics, then even in the absence of input, autonomous dynamics will encode a particular trajectory as the system relaxes to a fixed point. This trajectory can then be exploited by the learned optimizer to induce changes in optimization parameters, such as the effective learning rate.

---

[4]We run PCA across a large set of optimizer states visited during test examples to visualize optimizer state trajectories.

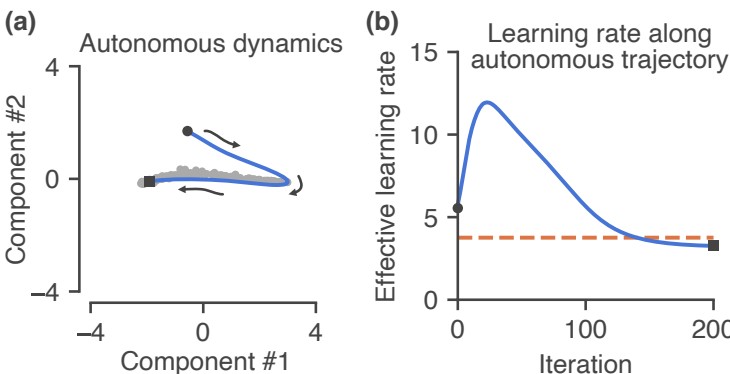

Figure 5: Learning rate schedules mediated by autonomous dynamics, shown for the linear regression task (additional tasks are in App. A.3). **(a)**: Low-dimensional projection of the dynamics of the optimizer in response to no input (blue line) around approximate fixed points (gray circles). These autonomous dynamics allow the system to learn a learning rate schedule (see §5.3). **(b)**: Effective learning rate computed at the autonomous state trajectories in (a). Dashed line indicates the best (tuned) learning rate for momentum on this task.

Indeed, we find that for two of the tasks (linear regression and MNIST classification) learned optimizers learn an autonomous trajectory[5] that modifies the learning rate, independent of being driven by any actual gradients.

This trajectory is shown for the linear regression task in Fig. 5a, starting from the initial state (black circle) and converging to a global fixed point (black square). Along this trajectory, we compute update functions and find that their slope changes; this is summarized in Fig. 5b as the effective learning rate changing over time. Results from the other tasks are presented in App. A.3.

## 5.4 Learning rate adaptation

The next mechanism we discovered is a type of learning rate adaptation. The effect of this mechanism is to decrease the learning rate of the optimizer when large gradients are encountered. The effect is qualitatively similar to adaptive learning rate methods such as AdaGrad or RMSProp, but it is implemented in a new way in learned optimizers.

To understand how momentum is implemented by learned optimizers, we studied the linear dynamics of the optimizer near a fixed point (§5.1). That fixed point was found numerically (§4.2) by searching for points $h^*$ that satisfy $h^* \approx F(h^*, g^*)$, where we hold the input (gradient) fixed at zero ($g^* = 0$). To understand learning rate adaptation, we need to study the dynamics around fixed points with non-zero input. We find these fixed points by setting $g^*$ to a fixed non-zero value.

We sweep the value of $g^*$ over the range of gradients encountered for a particular task. For each value, we find a single corresponding fixed point. These fixed points are arranged in an S-curve, shown in Figure 6a. The color of each point corresponds to the value of $g^*$ used to find that fixed point. One arm of this curve corresponds to negative gradients (red), while the other corresponds to positive gradients (green). The tails of the S-curve correspond to the largest magnitude gradients encountered by the optimizer, and the central spine of the S-curve contains the final convergence point[6].

These fixed points are all attractors, meaning that if we held the gradient fixed at a particular value, the hidden state dynamics would converge to that corresponding fixed point. In reality, the input (gradient) to the optimizer is constantly changing, but if a large (positive or negative) gradient is seen for a number of timesteps, the state will be attracted to the tails of the S-curve. As the gradient goes to zero, the system converges to the final convergence point in the central spine of Fig. 6a.

---

[5]Note that this autonomous trajectory evolves in a subspace orthogonal to the readout weights used to update the parameters. This ensures that the autonomous dynamics themselves do not induce changes in the parameters, but only change the effective learning rate.

[6]Fig. 6a uses the same projection as in Fig. 3a, it is just zoomed out (note the different axes ranges).

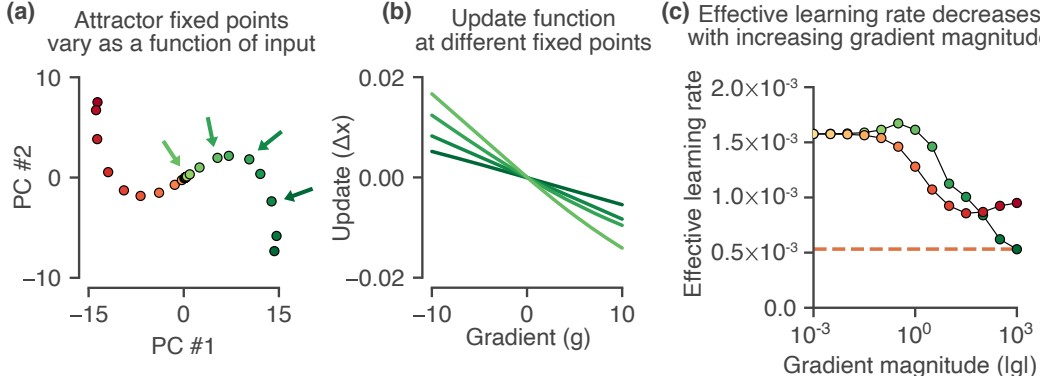

Figure 6: Learning rate adaptation in learned optimizers, shown for the Rosenbrock task (results are similar for other tasks, see App. A.4). **(a)** Approximate fixed points (colored circles) of the optimizer state dynamics computed for different gradients reveals an S-curve structure. Large positive (negative) gradients push the optimizer state to the dark green (red) tails of the S-curve. **(b)** Update functions (§4.1) computed at different points along the S-curve, corresponding to the arrows in (a). The effect of moving towards the tail of the S-curve is to make the update function more shallow (and thus have a smaller learning rate, cf. Fig. 2d). The effect is similar along both arms; only one arm is shown for clarity. **(c)** Summary plot showing the effective learning rate along each arm of the S-curve, for negative (red) and positive (green) gradients.

What is the functional benefit of these additional dynamics? To understand this, we visualize the update function corresponding to different points along the S-curve (Fig. 6b). The curves are shown for just one arm of the S-curve (green, corresponding to positive gradients) for visibility, but the effect is the symmetric across the other arm as well. We see that as we move along the tail of the S-curve (corresponding to large gradients) the slope of the update function becomes more shallow, thus the effect is to decrease the effective learning rate.

The learning rate along both arms of the S-curve are summarized in Fig. 6c, for positive (green) and negative (red) gradients, plotted against the magnitude of the gradient on a log scale. This mechanism allows the system to increase its learning rate for smaller gradient magnitudes. For context, the best tuned learning rate for classical momentum is shown as a dashed line.

## 5.5 Tuning per layer and parameter type

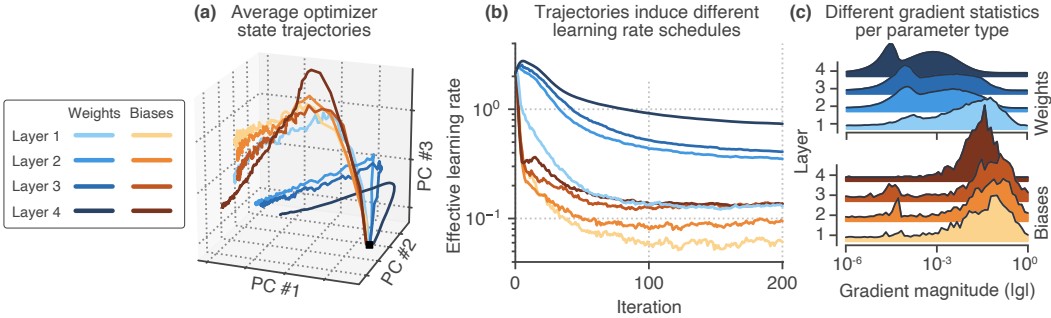

Figure 7: Parameter type-specific tuning found in a learned optimizer trained on an MNIST classification problem. **(a)** Trajectories of the optimizer, averaged across parameters within a particular layer and of a specific type (either weight or bias), exhibit different trajectories through optimizer state space. **(b)** Effective learning rate (computed as the slope of the update function) along each of the trajectories from panel (a). **(c)** Histograms of gradient magnitudes separated by layer and parameter type. The learned optimizer uses these differences in gradient statistics like these to induce the different trajectores from panels (a) and (b).

The final behavior we identified only exists in the learned optimizer trained on the MNIST CNN classification problem. It is a way for the learned optimizer to tune optimization properties (such as the effective learning rate) across different layers and across parameter types (either weight or bias) in the CNN being trained. We can see this most easily by taking all of the optimizer state trajectories for a particular layer or parameter type, and averaging them. These average trajectories are shown in Figure 7a projected onto the top three PCA components. The initial state is in the bottom right, and trajectories arc up and over before converging over to the left side of the figure.

What is the functional benefit of separating out trajectories according to parameter type? To investigate this, we computed the update function at each point along the trajectories in Figure 7a. We found that the effective learning rate varied across them (Fig. 7b). In particular, the bias parameters all have a much lower learning rate than the weight parameters. Within a parameter type, later layers have larger learning rates than earlier layers.

A clue for how this happens can be found by looking at the gradient magnitudes across layers and parameter types at initialization (Fig. 7c). We see that the bias parameters (in orange) all have much larger gradient magnitudes on average compared to the weight parameters (in blue) in the later layers. This is a plausible hypothesis for the signal that the network uses to separate trajectories in Fig. 7a.

We want to emphasize that these are correlative, not causal, findings. That is, while the overall effect appears as if the network is separating out state trajectories based on layer or parameter type, it is possible that the network is really attempting to separate trajectories based on some additional factor that happens to be correlated with depth and or parameter type in this CNN.

## 6 Discussion

In this work, we trained learned optimizers on four different optimization tasks, and analyzed their behavior. We discovered that learned optimizers learn a plethora of intuitive mechanisms: momentum, gradient clipping, schedules, forms of learning rate adaptation. While the coarse behaviors are qualitatively similar across different tasks, the mechanisms are tuned for particular tasks.

While we have isolated specific mechanisms, we still lack a holistic picture of how these are stitched together. One may be able to extract or distill a compressed optimizer from these mechanisms, perhaps using data-driven techniques [42, 43] or symbolic regression [44].

The methods developed in this paper also pave the way for studies of when and how learned optimizers generalize. By mapping different mechanisms to the underlying task used to train an optimizer, we can identify how quantitative properties of loss surfaces (e.g. curvature, convexity, etc.) give rise to particular mechanisms in learned optimizers. Understanding these relationships would allow us to take learned optimizers trained in one setting, and know when and how to apply them to new problems.

Previously, not much was known about how learned optimizers worked. The analysis presented here demonstrates that learned optimizers are capable of learning a number of interesting optimization phenomena. The methods we have developed (visualizing update functions and linearizing state dynamics) should be part of a growing toolbox we can use to extract insight from the high-dimensional nonlinear dynamics of learned optimizers, and meta-learned algorithms more generally.

### Acknowledgments

The authors would like to thank C. Daniel Freeman and Ben Poole for helpful discussions and for comments on the manuscript.

### Funding transparency

The authors were employed by Google, Inc. while this research was being conducted.

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
