# OpenReview forum: "Reverse engineering learned optimizers reveals known and novel mechanisms"
_NeurIPS.cc/2021/Conference — NeurIPS 2021 Poster_

### Official Review · Reviewer_Phij · 2021-07-15

**Rating:** 7
**Confidence:** 3

**Summary:**

The authors devise tools for investigating the behaviour of learned, gradient-based, optimization algorithms in order to understand better how they achieve their empirically validated high performance.

To this end, they visualize the update function, i.e., the change in parameters in dependence of the gradient input, which allows to identify mechanisms employed also by hand-designed optimizers, such as gradient clipping, momentum, and learning rate adaptation. The authors demonstrate that this visualization is particularly insightful when analyzing the update rule using the linearized optimizer state dynamics around fixed points. Furthermore, the authors identify that learned optimizers encode learning rate schedules through autonomous state trajectories.

Using their tools, the authors analyze one existing type of learned optimizer (the RNN-based approach proposed by Andrychowicz et al., "Learning to Learn by Gradient Descent by Gradient Descent") on four relatively simple problems (linear regression, Rosenbrock, two moons, MNIST). They find that (depending on the problem) the learned optimizer can exhibit well-known mechanisms like gradient clipping, momentum, learning rate adaptation, and learning rate schedules. Furthermore, they provide clues that the optimizer can learn parameter-type (bias/weights) and layer-specific learning-rate schedules in neural-network based problems.

**Ethical Concerns:**

No ethical concerns.

**Limitations And Societal Impact:**

No potential negative societal impact.

**Main Review:**

Originality: The authors propose to combine visualization tools and methods from dynamical system analysis to gain useful insights into the behaviour of learned, gradient based optimizers. While neither their analyzation methods nor the analyzed optimization algorithm are novel, the application of those methods to better understand the learned optimizers is new and the way the authors combine said methods is clever and insightful.

Quality: The proposed analyzation approach uses well-understood methods from dynamical system analysis to visualize the behaviour of learned optimizers. The authors apply these methods in a technically sound manner and on a range of simple problems. As the approach is intuitive and concise, it yields a range of useful insights into the behaviour of learned optimizers and unveils both parallels to known optimization mechanisms as well as novel behaviours. The authors admit that their work does not yet yield a "holistic picture" of learned optimizer behaviour. I agree with that, i.p., because the authors analyze only one particular type of learned optimizer. The submission could be strengthened by including different learned optimizer architectures.

Clarity: The submission is very well written and the visualizations are of high quality. In my opinion, the exposition could be strengthened by considering the following points:

- Sec. 5.1:
    - Maybe try to merge some of the information/equations of App. C into the main text.
    - Provide some more background on why the "learned optimizer uses a single eigenmode to implement momentum" (l. 189).
- Sec. 5.3:
    - Provide some more background on the connection between autonomous trajectories and the implementation of learning rate schedules.
- Sec. 3.2:
    - How do you generate a distribution of problems for MNIST? Do you also sample the initial weights of the CNN?

Minor points (not influencing my score):

- l. 36: three disparate tasks → four disparate tasks
- Eq. (2) does not contain the readout function $r$ referred to in l. 658.

Significance: The authors compile a toolbox of analyzation methods for learned optimizers. While the used methods are not new, the authors apply them in an intuitve and clever way. Similarly, while the experiments do not yet yield a holistic picture (i.p. because the authors apply their tools to only one type learned optimizer), they still provide some interesting and important insights into the behaviour of the RNN-based optimizer. Therefore, I think the submission in its current form is significant, i.p., because future work will benefit from it. I therefore recommend acceptance.

---------------------------------------------------
After reading the other reviews, I am even more convinced that the paper is interesting and valuable and the proposed methodology is sound. In addition, the authors will improve on the clarity of their manuscript. Thus, I raise my score to 7.

**Time Spent Reviewing:**

4

---

> ### Author Response · Authors · 2021-08-09
> **Thank you for the suggestions on improving the clarity of our work!**
>
> We wish to thank the reviewer for their time and for the great suggestions on how to improve the clarity of our work.
>
> *Steps taken to improve clarity*: We have amended the manuscript to include more information from Appendix C, and expanded on the statement that the learned optimizer uses a single eigenmode to implement momentum. We have additionally added context on the connection between autonomous trajectories and hyperparameter schedules.
>
> *Generating a distribution of problems for MNIST*: Yes, that’s correct! The distribution comes from sampling initial weights of the CNN. We have updated the description of the MNIST problem to make this clear.

---

### Official Review · Reviewer_QfSG · 2021-07-16

**Rating:** 7
**Confidence:** 4

**Summary:**

This submission focuses on analyzing the behaviors of learned optimizers. The goal is to elucidate properties and patterns learned by such learned optimizers, in order to better analyze, understand, and lay groundwork to potentially improve the behavior of learned and/or manually-designed optimizers. A novel reverse engineering approach is taken and identifies learned interpretable and fundamental behavior such as momentum, gradient clipping, learning rate schedules, and learning rate adaptations.

The submission trained learned optimizers on each of four representative tasks: Linear Regression, Rosenbrock, Two Moons, MNIST . These tasks were selected because they are fast to train, cover a range of loss surfaces (convex and non-convex, low- and high-dimensional, deterministic, and stochastic). Additionally, three baseline optimizers (momentum, RMSProp, and Adam) were tuned individually for each task. The learned optimizers outperformed the baseline optimizers on the meta-objective.

The submission uses two primary methods to analyze optimizer behavior and mechanisms: 1) visualize optimizer update function at a particular optimizer state. 2) analyze optimizer dynamics around fixed points.  The study was able to show that learned optimizers implement classical momentum and a soft form of gradient clipping. The paper visualizes and provides a careful explanation for the best learned optimizer for Rosenbrock (others in appendix) for both. The report also analyzes learned behavior for learning rate schedules and learning rate adaptation. The study isolated the behavior of tuning per layer and parameter type in the CNN for the MNIST problem.

The submission includes a detailed Appendix that illustrates the same behaviors for all the other learned optimizers not included in the main paper, along with other materials such as the meta-learning approach.


**Ethical Concerns:**

No ethical concerns.

**Limitations And Societal Impact:**

No potential negative societal impact.

**Main Review:**

The topic of the submission, learned optimizers, is an important and timely topic. Overall, the submission is well-written, takes a systematic approach to analyzing the behaviors of learned optimizers, and leverages visualizations and detailed explanations to provide evidence of the learned behaviors.  The findings are useful in validating that behaviors of learned optimizers align with manually-designed optimizers, and equally important, valuable as a foundation for further analysis and potential advancement of learned optimizers.

The cases for momentum, gradient clipping, and learning rate adaptation were especially clearly laid out and visualized.

The learning rate schedule discussion (Section 5.3) illustrated that a learned optimizer adjusted the learning rate over iterations. The report claims the learned optimizer is implementing a “schedule” using autonomous (not input driven) dynamics. However, it is unclear from this analysis that the learned optimizer is actually implementing a “schedule” or has only learned to adapt the learning rate. Additional analysis and evidence to support the assertion that a schedule was learned would strengthen this section.

The learning rate adaptation discussion (Section 5.4), explained well the effects of the learning rate adaptation by the learned optimizer. The illustrations breaking down the behaviors for the positive and negative gradients were well explained and very helpful for illustrating adaptation taking place by the optimizer.  However, line 236 refers to a “new mechanism” implemented by the learned optimizer. Given that manually-designed/baseline optimizers also leverage adaptive learning rates, it was unclear why this is being characterized as a “new mechanism.”

The additional materials and discussion in the Appendix materially added to the findings, and it would have been helpful to have had some of the information from the Appendix (e.g., potential functional benefits of some of the learned behaviors) earlier, in the main text).

It would also be interesting to understand which of the identified behaviors the learned optimizers that performed suboptimally did not exhibit and whether some connection could be made to these behaviors and the performance of the optimizer.

Nit: 193 “(gray line in Fig. 3b)” Perhaps it was an issue with the rendering, but I could not see a gray line in 3b (?)






**Time Spent Reviewing:**

2 hours

---

> ### Author Response · Authors · 2021-08-09
> **Thank you for your review! Comments and clarifications addressed below.**
>
> We wish to thank the reviewer for their time and for providing feedback on our manuscript!
>
> *Autonomous dynamics implementing a schedule*: The reason we refer to the autonomous dynamics as implementing a schedule, is that the autonomous dynamics are the dynamics of the optimizer when there is no gradient input to the optimizer (that is, the input gradients to the optimizer are all zero). Thus, the learned optimizer can not be adapting to the gradients in this case, since the gradients are identically zero. Thus, when the learned optimizer changes some hyperparameter (such as learning rate) in the absence of gradient input, then we treat that as a schedule since it happens independently of the input.
>
> *Referring to the learning rate adaptation as a "new" mechanism*: We agree that this language is confusing, and have clarified this point in the text. The reason we refer to this as a “new mechanism” is because the particular way this adaptation is implemented in the learned optimizer is different from how adaptive learning rates are typically implemented in hand-designed optimizers. In hand-designed optimizers, learning rate adaptation is implemented by normalizing gradients by an estimate of the norm (or second moment) of recent gradients. Here, we find that learned optimizers implement adaptive learning rates through different approximate fixed points in the dynamics, each of which has a different effective learning rate.
>
> We have updated the text to refer to this as a “new way of implementing adaptive learning rates”. This also suggests that future work studying this implementation in detail may shed light on if or how this new method for implementing adaptive learning rates could be superior to using second moment estimates.
>
> *Additional material in the Appendix*: We have revised the manuscript and brought in some additional detail and discussion about the learned optimizer mechanisms from the Appendix into the main text.

---

> > ### Comment · Reviewer_QfSG · 2021-08-25
> > **Response**
> >
> > Dear authors,
> > I have read your response. You have addressed the concerns raised in my review.
> >
> > Thank you.

---

### Official Review · Reviewer_sWNV · 2021-07-17

**Rating:** 6
**Confidence:** 4

**Summary:**

The paper deals with the analysis of the learned optimizers (LO). In particular, it tries to isolate their properties and distinctions with respect to the existing baseline optimizers.

It is true that a nascent field of learned optimizers, while enjoying good performance, currently lacks in depth analysis of why the results work. This paper doesn't answer all the questions, but it attempts to bridge the gap between a mysterious performance of LO and known well analyzed optimization techniques like momentum, gradient clipping, LR schedules and adaptation.

The paper uses 4 different tasks for the analysis (Linear regression, Rosenbrock, Two moons and MNIST) and uses a simple RNN optimizer for a LO.

The presented empirical result is analyzed from a theoretical perspective of a general LO-state update function. The theory has a very mild assumptions on the LO for analysis (essentially only that they have to converge to the fixed point).

The paper is mostly well written, however, some parts are hard to read, misleading or not explained well (see bellow).

**Limitations And Societal Impact:**

The authors did not include any section to address societal impact of their work. Since this is mostly an analysis of _existing_ algorithms, I think it is ok.

**Main Review:**

First of all, I think that these kind of analysis paper are very valuable and I encourage the authors to design a toolbox that can be applied to a given LO in order to analyze and understand its performance.

My main concern is the generalization and applicability of the analysis of the paper. From what I understand the paper interlaces and switches between two modes:
- theoretical analysis of a general LO of the form from eqn. (1-2). This was used e.g. to derive an expression for Momentum in section 5.1.
- empirical analysis of 4 tasks with limited theoretical understanding (e.g. section 5.2, 5.3 and partially 5.4).

The theoretical analysis has a very mild assumptions on the LO form and only requires that the optimizer state converges to a fixed point. The empirical analysis focuses on the RNN LO, but I'm curious what example of a general convergent update rule *are not* covered by this analysis? Is RNN just use as a general example with any other LO being applicable as well?

It is a little bit confusing how much we can really generalize beyond 4 tasks and which conclusions to make out of this analysis? The authors showed that a given optimizer exhibit certain types of behavior that we know from known optimizers, but _is this surprising and unexpected_? What can we learn from this analysis, beyond just "it is a first step and we hope the results would be find useful down the line"? It would be great if the authors add a comparative table in the main text (or appendix) with a summary of what they learned from each of 4 tasks. Together with a loss decrease curves from fig 1 top these would be a good succinct outcome.

Most of the other comments below are just clarifying the confusions when I read the paper. While this is not crucial, the paper would benefit greatly from a careful proofreading.

Comments:
- I would be a little careful with the notation, e.g. l.61 defines $w$ as readout weights. I don't see why they are called like this? What exactly do they represent? They are not defined in Andrychowicz et al nor anywhere else I looked. Are $w$ are being used at all in the paper?
- l.69: state variable is a scalar, later: state var are low-dimensional.
- l.72: "dynamics are straightforward". Arguably so, e.g. Nesterov accelerated gradient.
- a lot of important information is hidden in the Appendix. It is quite hard to read the paper as is. without looking back and forth to the Appendix.
- l84: "fast to train". Not precise enough. Fast with respect to what? Low number of iterations? Fast gradient computation? All of the above?
- l.86: maybe instead of random, create 2-3 group with low/normal/hard Hessian conditioning? I don't think that fig 1(a) would be too packed if one adds few more curves.
- Fig 1: maybe use log-x and show the dynamic till convergence?
- Fig 1: different shares of red/orange are very hard to distinguish. Consider using different color palette.
- which tasks were used to train the meta-optimizer? Same as the training? The paper mention this at l.164, but it is not clear.
- l.178: misleading: learned optimizer is not *approximately linear*, the authors *linearly approximate it* using the Jacobian. These two statements mean very different things.
- Momentum is defined differently at l.135 than at line 623 and uses different lingo (velocity and mom. hyperparameter vs timescale and learning rate).

**Time Spent Reviewing:**

8

---

> ### Author Response · Authors · 2021-08-09
> **Thank you for your review! Comments addressed below.**
>
> We wish to thank the reviewer for their time and careful reading of our manuscript!
>
> *Learned optimizer architectures covered by our analysis*: Yes, our fixed point analysis is applicable to any learned optimizer that converges to a fixed point. Moreover, the visualization techniques are applicable to any update rule, even those that do not converge. We will make this clearer in the text.
>
> *Is this surprising or unexpected?*: First, we want to reiterate that we analyzed learned optimizers trained separately on each of the four tasks. Moreover, the four tasks chosen cover different types of optimization problems: convex and non-convex surfaces, stochastic and deterministic problems, low- and high-dimensional parameter spaces. We find it pretty remarkable that even when trained on such disparate tasks, learned optimizers appear to learn very similar mechanisms--this suggests that these mechanisms are likely to be useful across many other optimization tasks. It could have turned out that optimizers learned to use strategies that were highly task specific (mechanisms that were specific to an individual loss surface or dataset). Instead, we find that the mechanisms are general. Moreover, these mechanisms match or mimic known mechanisms in the optimizer literature. This suggests an optimistic outlook for learned optimizers: we can both trust the mechanisms that they learn, and use learned optimizers as a tool to study or identify new optimization strategies.
>
> *What can we learn from this analysis?*:  We draw the following conclusions from our analysis:
> - Learned optimizers learn intuitive and known optimization mechanisms, which gives confidence in deploying learned optimizers themselves as well as provides additional support for the utility of these mechanisms in hand-designed optimizers. If a meta-learning algorithm rediscovers a known mechanism, without any built in prior ability to do so, that provides additional evidence that the mechanism is probably a good idea to use when building optimization algorithms.
> - While the qualitative mechanisms are consistent across tasks, the quantitative aspects of the mechanisms vary between tasks. This suggests that in order to get learned optimizers to generalize, we need ways of disentangling the qualitative mechanism from its quantitative hyperparameters (as is done with hand-designed optimizers).
> - While the mechanisms are largely consistent across tasks, there are minor differences. We will add a table summarizing these, following your good suggestion. Characterizing the mechanisms that are used for a particular task gives us a new way to study or classify properties of different loss landscapes, and how properties of the loss surface influence optimization algorithms.
>
> *Additional minor concerns*: We greatly appreciate the careful proofreading of the manuscript. We have addressed all of the minor comments and clarifications suggested by the reviewer.

---

> > ### Comment · Reviewer_sWNV · 2021-08-27
> > **Generalization of the analysis**
> >
> > Thanks to the authors for the detailed answer to my questions!
> >
> > I've noticed that the authors almost always (title, body and even the review response) are using the term "learned optimizers" in the plural, whereas in reality they propose the analysis only of one learned optimizer, which is RNN based one from Andrychowicz et al. I would make it really clear very early in the text that in fact the authors are analyzing learned optimizers on a _single example of RNN optimizer_. Currently, it is located _very_ deep in the text and not mentioned in the abstract or even introduction or discussion. To a reader, this is very confusing. It might be true that the analysis might be applicable to other algorithms, but the paper only proposes techniques to do so without statements of generalizability. To that extend, even the title of the paper is quite misleading: "known and novel mechanisms" are revealed only for the case of RNN optimizer. The better title might be "Techniques for reverse engineering learned optimizers", since from what I understand this is closer to what the paper is actually proposes.
> >
> > In the response, the authors try to generalize the findings on 4 tasks from 1 optimizer and make a wide claim that learned optimizers use "general mechanisms" and that they don't "use strategies that were highly task specific". I would encourage the authors to very prominently state that the findings are provided for a very specific class of learned optimizers.
> >
> > The way I see it, this paper would be a great asset for benchmarking and initial analysis of a novel or existing learning optimizers techniques. It would be awesome to have a pluggable component that allows the practitioner to plug their LO and a given task and get a report on the how much their method is using momentum, clipping etc. _Ideally_, it should also provide a reverse analysis, suggesting that the LO relies too much of the task knowledge and therefore might be less applicable to other tasks.
> >
> > Based on the other reviewers responses and my own understanding of the paper, the technical contribution of the paper is evident and undeniable. However, the pivot in the narrative, as I underlined above, would be very important for a reader to have a better understanding of what this paper is about and its contribution.

---

> > > ### Author Response · Authors · 2021-08-30
> > > **Re: applicability across optimizer architectures**
> > >
> > > Thanks for the comment!
> > >
> > > We do not want there to be any confusion or misunderstanding regarding claims made in the title/abstract/intro and whether or not the results are sufficient to back up these claims. To that end, we can understand the reviewer's point regarding use of language such as "learned optimizers" (plural), "general mechanisms", etc.
> > >
> > > We do want to make the distinction that while we focus on one learned optimizer _architecture_, we study four _different_ sets of trained weights, trained on each of the four tasks. When we say that the "mechanisms are general", we mean that they generalize across the different sets of trained weights (across the tasks). We apologize for the confusion! We have gone through the paper and updated these statements to make it clear that we mean the mechanisms generalize across the training tasks, but are all using the same optimizer architecture. Finally, we are on board with changing the title to reduce confusion (we do like "Techniques for reverse engineering learned optimizers" as an option).
> > >
> > > The question of how different learned optimizer architectures (besides RNNs) constrain, change, or modify observed mechanisms is definitely one that we are interested in--we have updated the discussion to discuss this important direction for future work.
> > >
> > > To summarize: we see your point, and have changed the text throughout the paper (but especially in the intro and abstract) to make it clearer that we are focused on generalization across training tasks, for the same optimizer architecture (as opposed to generalization across architectures). We will also change the title, per the reviewer's suggestion.

---

> > > > ### Comment · Reviewer_sWNV · 2021-08-30
> > > > **Re: applicability across optimizer architectures**
> > > >
> > > > Great. Happy to hear that my comments are well received :)
> > > > Slight change of narrative to be more precise would go a long way to help people understand the contributions better.
> > > >
> > > > Comments that I still have (mostly a wish list of things I would love to see in the paper):
> > > > - I would like to have a discussion on the sanity check of the proposal: is it guaranteed that if one plugs momentum optimizer as a LO example, it would completely recover momentum and would show no signs of clipping etc? Is that guaranteed for _any_ task? I'm not 100% this is true, given the empirical nature of the most of the analysis.
> > > > -  I would love to see an example when LO _does not_ mimic existing optimizers and actually memorizes the task without exhibiting the properties of the known optimizers. What would the analysis show in that case?
> > > > - The momentum analysis (sec 5.1 and App C) relies on the linear approximation of the recurrent dynamics. It is not clear to me that this approximation is useful and actually holds in practice. It could be helpful to actually run the momentum optimizer with the parameters recovered by this analysis and see if it is close to the training dynamics of the LO. Of course, this also would be not accurate, since the LO does much more than just momentum...

---

> > > > > ### Author Response · Authors · 2021-09-01
> > > > > **Response to additional comments**
> > > > >
> > > > > Hi,
> > > > >
> > > > > Great questions/comments. Responses below:
> > > > > - This is a great sanity check, and empirically, we are fairly confident that you would exactly recover things like momentum if that was the optimizer used (as opposed to a learned optimizer). This depends on the optimization problem used to generate the data for reverse engineering, but it would be difficult to come up with a problem such that you would _not_ recover momentum. One reason for this is that each coordinate or parameter in the optimization problem is effectively like seeing another sample/snapshot of the momentum dynamics, so for optimization problems with more than a few variables you effectively already have plenty of data needed to observe that the dynamics are linear (as they are for momentum). Another reason for this is that the data are noiseless (up to numerical precision), since the data are generated in the computer (note that this is true even for stochastic problems, since the stochasticity is in the inputs/gradients, not the dynamics). This means that reverse engineering optimizers is a much easier than, say, fitting a dynamical system to data recorded from a physical process in the natural world (which will be noisy).
> > > > > - Another great question. We have not come across a LO that does not mimic existing optimizers in our experiments, but it is an interesting question to try and think about how one might change how the LO is trained in order to get it to do something different or more exotic. Perhaps one could train a LO in some kind of restricted setting, or limit its access to problem information, or somehow pick specific optimization problems that induce interesting behavior in the LO.
> > > > > - Yes, another great sanity check. This analysis is actually in Appendix B, where we analyzed a learned optimizer trained on the quadratic task. Here, the reverse engineering analysis reveals that this particular learned optimizer recovers momentum exactly. In fact, the actual eigenvalues of the LO match exactly the optimal learning rate and momentum hyperparameter for this problem distribution (which we can calculate exactly since the optimization problem is quadratic). This is shown in Appendix Figure 14. You can see in Fig. 14A that there is one eigenmode that dominates the dynamics, and this eigenmode has a corresponding learning rate and momentum value that matches the optimal one in Fig. 14C. Finally, if we replace the LO with a rank-1 approximation (that is, we kill the other eigenmodes in the LO) then we do not see any change in performance (which confirms that the LO is only using that one eigenmode).
> > > > >
> > > > > Thank you for the great comments!

---

### Official Review · Reviewer_WPbX · 2021-07-23

**Rating:** 6
**Confidence:** 3

**Summary:**

In this work, the author got trained learned optimizers on four different optimization tasks, and analyzed their behavior. They discovered that learned optimizers learn a plethora of intuitive mechanisms: momentum, gradient clipping, schedules, forms of learning rate adaptation. While the coarse behaviors are qualitatively similar across different tasks, the mechanisms are tuned for particular tasks.

**Ethical Concerns:**

No ethical concerns

**Limitations And Societal Impact:**

This work give visualizition of the learned optimizer. But they give very limited suggestions about how to benefit from it. Such as how to design a better optimizer? How to select the design between different tasks and so on.

**Main Review:**

Originality:
To the best of my knowledge, this kind of work which try to figure out the working mechanism of learned optimizer is rare. Very few people try to interpret it but they want to put forward a powerful optimizer. Thus, I think this work is innovative.

Quality:
I have seen that you do experiments on different suitations. This is very good and I believe that your analysis is convincing.

Clarity:
The description and visualizition part is good.

Significance:
I admit that your work is innovative and give new idea to the following researchers. However, after visualizition of the learned optimizer. You give very limited suggestions about how to benefit from it. Such as how to design a better optimizer? How to select the design between different tasks and so on.

**Time Spent Reviewing:**

0.5 hours

---

> ### Author Response · Authors · 2021-08-09
> **Regarding the significance of work on understanding learned optimizers.**
>
> First, we wish to thank the reviewer for their time and for providing feedback on our manuscript!
>
> Second, regarding the significance of our work. Our work benefits the learned optimizer community because it tackles a fundamental question about learned optimizers: how do they work, and do their mechanisms generalize across tasks and datasets? We see our work as an important first step in building out the science of understanding learned optimizers, and that the fundamental understanding provided by our work is itself of interest.
>
> That being said, we will provide additional examples of ideas for improving optimizers as a result of our work in the discussion. For example, one finding of our work is that to get learned optimizers to generalize better across tasks, we need to build architectures that can separate the general mechanism (such as learning rate adaptation, momentum, etc.) from the quantitative values used for that mechanism (the specific learning rates, momentum timescales, etc.); since across tasks the mechanisms generalize, but the quantitative values do not. Though it is it’s own research project, one concrete future direction would be to hard code the mechanisms we found in the RNN optimizers, and then meta-learn parameterizable schedules for these mechanisms. Such a study would validate (or reject) the mechanisms found in the work here. This is just one example of how better understanding of what learned optimizers are doing can lead to improvements.
>
> We have updated the discussion section with this and additional take home messages resulting from our work.

---

### Author Response · Authors · 2021-08-09
**General response to all reviewers**

We thank the reviewers for their time, careful reading of our manuscript, and insightful feedback. We are pleased that all the reviewers praised the novelty of our work, and called out the insights provided by our methods. Reviewers observed that the work “is innovative”, that  “the way the authors combine said methods is clever and insightful”, and commented that “this kind of work … is rare”, and that it lays a “foundation for further analysis and potential advancement of learned optimizers”.

Specific concerns and suggested improvements are addressed in responses to individual reviewers below.

---

### Decision · Program_Chairs · 2021-09-27

**Decision:**

Accept (Poster)

**Comment:**

This paper was very actively discussed amongst the reviewers.

Points in favor were summarized by reviewer QfSG as follows:

1. The analytical tools presented in the paper are a strong foundation for characterizing learned optimizers. Such tools are lacking in the field. In addition to the tools themselves (which are useful for other L2O researchers), the systematic approach taken to the experiments and explanations could also be helpful for others seeking to contribute such tools (for and beyond L2O methods).

2. The methodology, experiments, and conclusions taken from the experiments are well-grounded and solid, especially if one takes into consideration the significant additional depth provided in the appendix.

3. The results show that learned optimizers match or mimic techniques in state-of-the-art optimizers, despite being trained on diverse types of optimization problems - which gives merit to both L2O approaches and to the tools presented. That is, even if it is not surprising, given how learned optimizers are derived, that learned optimizers learn similar techniques as known optimizers -- that finding (learning to similar methods) brings merit to L2O approaches as finding 'trusted' approaches.

4. The authors demonstrates applicability beyond just the 4 tasks, and responded to the concerns.

Reviewer sWNV replied to these 4 points as follows:

1. I agree that these tools are lacking and having a paper like this (and especially a toolbox) would lie a good foundation onto a better understanding of the learned optimizers. This is undeniable contribution and I'm happy that this paper tackles it.

2. The techniques provided in the paper are good, however, there are still some questions remain. For example, the momentum analysis (sec 5.1 and App C) relies on the linear approximation of the recurrent dynamics. It is not clear to me that this approximation is useful and actually holds in practice. It would be useful to actually run the momentum optimizer with the parameters recovered by this analysis and see if it is close to the training dynamics of the LO. Of course, this also would be not accurate, since the LO also do much more than just momentum...

3. I think that it is a bit of an overstatement to say that LO match or mimic known optimizers, they specifically being tested to match or mimic the specific property of a given optimizer. I would love to see an example when this does not happen and the LO solely memorizes the task without exhibiting the properties of the known optimizers.

4. My main criticism is that the from the title, introduction and the discussion it seems like the analysis holds and applies to many Learned Optimizers, whereas in reality it is only 1 that is being analyzed. This by itself is not a problem and doesn't deny the technical contribution, but it is misleading to the reader (especially the cursory reader that relies of Abstract + Intro for the quick understanding of the paper). I would really encourage the authors to pivot the narrative of the paper and instead focus on the techniques and methods of general LO analysis instead of claims about Learned Optimizers in general. To this extend, I would like to re-iterate that the release of the general and easy-to-use toolbox that can generate an extensive report given a used provided LO and a task (ideally a benchmark of many tasks) should be an essential part of this paper.

After this internal discussion, the reviewers offered to change the narrative and title of the paper to alleviate the concerns in 4. above.

Overall, the paper presents a solid contribution to the field of learned optimizers, and I recommend acceptance. I encourage the authors to act on the suggestions by reviewer sWNV, especially adapting narrative + title.